# Texture Interpolation for Probing Visual Perception

**Jonathan Vacher**[*]
Albert Einstein College of Medicine
Dept. of Systems and Comp. Biology
10461 Bronx, NY, USA
`jonathan.vacher@ens.fr`

**Aida Davila**
Albert Einstein College of Medicine
Dominick P. Purpura Dept. of Neuroscience
10461 Bronx, NY, USA
`adavila@mail.einstein.yu.edu`

**Adam Kohn**   **Ruben Coen-Cagli**
Albert Einstein College of Medicine
Dept. of Systems and Comp. Biology, and
Dominick P. Purpura Dept. of Neuroscience
10461 Bronx, NY, USA
`adam.kohn@einsteinmed.org`
`ruben.coen-cagli@einsteinmed.org`

## Abstract

Texture synthesis models are important tools for understanding visual processing. In particular, statistical approaches based on neurally relevant features have been instrumental in understanding aspects of visual perception and of neural coding. New deep learning-based approaches further improve the quality of synthetic textures. Yet, it is still unclear why deep texture synthesis performs so well, and applications of this new framework to probe visual perception are scarce. Here, we show that distributions of deep convolutional neural network (CNN) activations of a texture are well described by elliptical distributions and therefore, following optimal transport theory, constraining their mean and covariance is sufficient to generate new texture samples. Then, we propose the natural geodesics (*i.e.* the shortest path between two points) arising with the optimal transport metric to interpolate between arbitrary textures. Compared to other CNN-based approaches, our interpolation method appears to match more closely the geometry of texture perception, and our mathematical framework is better suited to study its statistical nature. We apply our method by measuring the perceptual scale associated to the interpolation parameter in human observers, and the neural sensitivity of different areas of visual cortex in macaque monkeys.

## 1   Introduction

**Texture synthesis** Among existing texture synthesis algorithms [41], few have been inspired by visual neuroscience and visual perception. One theory assumed that there exists a fundamental perceptual feature that composes a texture, termed "texton". Despite being falsified, this led to the formulation of the theory of stationary Gaussian textures, which can be obtained by phase randomization [15, 53]. A complementary view, inspired by the primary visual cortex (V1) [25], is that textures perception only depends on the statistics of the wavelet coefficients of the texture (wavelets can be viewed as a standardized collection of "textons" [26]). An implementation of this hypothesis allows for the synthesis of textures by matching the marginal statistics (histograms) of the wavelet coefficients of a white noise image to those of a texture example [22, 6]. Pursuing this idea, Portilla and Simoncelli (PS) obtained high quality new texture samples by iteratively matching a set of higher-order summary statistics of the wavelet coefficients [39, 54]. The PS approach has also been successful in synthesizing sound textures [33]. These algorithms have been further improved with a proper mathematical framework to ensure convergence [49, 50]. A more recent approach uses deep learning to synthesize high-quality textures [16]. This approach is similar to PS [39], but instead of

---

[*]JV is now based at Laboratoire des Systèmes Perceptifs (LSP), Département d'études cognitives, École Normale Supérieure, PSL University, 75005 Paris, France

wavelet coefficients it consists in matching the statistics of a pre-trained CNN's activations of white noise to those of a texture example. Despite this progress, it remains unclear what computational ingredients are necessary for texture synthesis [52] (e.g., training the CNN weights may not be necessary [21, 52]). Lastly, although our focus here is on texture synthesis approaches that are related to visual perception, there are also multiple approaches that use CNNs, focusing on other aspects such as mathematical methods [31] and computer graphics [51, 62, 64].

**Perception and neural encoding of textures** Gaussian textures [15] and PS textures [39] have been widely used to study human visual perception (see [56] for a review). Gaussian textures are useful because they can be well parametrized [53] to answer specific questions, *e.g.* related to orientation and spatio-temporal frequency content of images [30, 20]. However, they lack the natural complexity that is captured by PS textures [39]. For this reason, PS textures, often called "naturalistic", have been instrumental to understanding image processing in the visual cortex beyond area V1 [14, 35, 65, 66, 36]. The PS texture model also accounts for various aspects of visual perception including categorization [2], crowding [3] and visual search [44], and has been proposed as a general model of peripheral vision [13, 43] despite some limitations [58, 60, 24]. However, different from simple Gaussian textures, PS textures cannot be easily modeled, and it is difficult to identify perceptually relevant axes in the space of PS summary statistics. Indeed, these statistics consist of a list of heterogeneous features (*i.e.* skewness, kurtosis, magnitude and phase correlations) which are not comparable, and prevent the use of simple closed-form probabilistic descriptions. To circumvent this problem, CNN texture synthesis [16] algorithms are promising because: (i) they can synthesize naturalistic textures that are perceptually indistinguishable from their original version [59]; (ii) they have a simple description based on homogeneous features (deep network activations at each layer) characterized by their mean and covariance, which will allow for more practical modeling.

**Texture interpolation and optimal transport** Texture interpolation or mixing is a niche in the broader field of texture synthesis. It consists of generating new textures by mixing different texture examples. To our knowledge, PS texture interpolation (and its equivalent for sound textures) has been used in only few neurophysiology and perceptual studies [14, 35, 34], relying on ad-hoc interpolation methods. Instead, texture interpolation is of main interest in computer graphics [4, 45, 5] and to illustrate optimal transport (OT) algorithms [42, 61]. Recent work combines Gaussian models [61] and CNN-based texture synthesis to perform texture interpolation [63]. Other related work further formalizes this, using a statistical description of textures. This is because interpolation between two textures naturally arises as their weighted average, which, could be properly defined in a statistical framework. One approach is to use OT [38] which gives a geometry to the space of probability distributions. Together with the hypotheses that perception of textures is statistical [56] and, that the brain represents probability distributions [40], the OT framework appears as a highly appropriate and unifying framework to study texture perception and its neural correlates. Specifically, it allows for the fine exploration of the space of natural textures by generating and using texture samples along a 1 dimensional geodesic as stimuli.

**Contributions** Our work provides several contributions that could serve vision studies (Figure 1), which we illustrate using the VGG19 CNN [48]. First, we show that CNN activations of natural textures have elliptical distributions (distribution with elliptical contour lines), which can be described by their mean and covariance even though they are not Gaussian. Leveraging OT theory [38], we show that enforcing these statistics corresponds to matching their distributions. By exploiting the natural geodesics arising with the OT metric, we define the interpolation between arbitrary textures. We compare interpolations obtained with alternative methods, and argue that ours is the most relevant to study the statistical nature of texture perception and its neural basis. Our results also suggest that training the CNN is necessary to generate interpolations that respect perceptual continuity. Lastly, we demonstrate how to use texture interpolation for human psychophysics and monkey neurophysiology experiments. In psychophysics, we use Maximum Likelihood Difference Scaling (MLDS [29, 32]) to measure the perceptual scale of the interpolation weight (*i.e.* the position of the interpolated texture on the geodesic joining two textures). We find that the perceptual scale is reliable for individual participants, and across texture pairs it ranges from linear to threshold non-linear. In neurophysiology, we study the tuning of visual cortical neurons in areas V1 and V4 to interpolation between a naturalistic texture and a spectrally matched Gaussian texture. Using population decoding, we find that adding naturalistic content to the Gaussian texture (*i.e.* moving along the geodesic towards the natural image) does not increase the stimulus-related information in V1, while it increases linearly with the interpolation weight in V4. We provide code to perform texture synthesis and

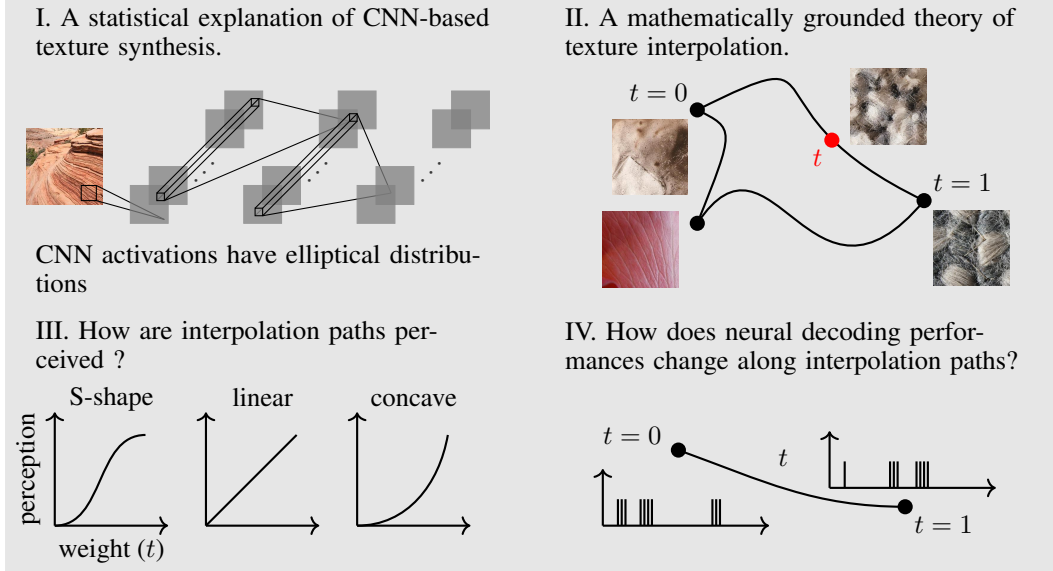

I. A statistical explanation of CNN-based texture synthesis.

CNN activations have elliptical distributions

III. How are interpolation paths perceived ?

II. A mathematically grounded theory of texture interpolation.

IV. How does neural decoding performances change along interpolation paths?

*Figure 1: Outline of our contributions.*

interpolation that can be run using a simple command line on a computer with a nvidia GPU or CPUs only[1]. We also provide the code to run the experiments both using psychtoolbox [28] and jspsych [9].

## 2  Methods

In the following paragraphs, first we define the class of elliptical distributions that we hypothesize are a good description of CNNs activations to natural textures. Then, we propose a method to quantify the elliptical symmetry of a high-dimensional, empirical distribution, to test our hypothesis. Next, we briefly define the OT framework and apply it to elliptical distributions. We then describe our framework for texture synthesis and interpolation.

**Elliptical distributions** A random vector $X \in \mathbb{R}^D$ ($D \in \mathbb{N}$) is elliptically distributed [19] if its density $\mathbb{P}_X$ can be written as

$$\mathbb{P}_X(\mathbf{x}; \boldsymbol{\mu}, \boldsymbol{\Sigma}, g) = c_n |\boldsymbol{\Sigma}|^{-\frac{1}{2}} g((\mathbf{x} - \boldsymbol{\mu})^{\mathrm{T}} \boldsymbol{\Sigma}^{-1} (\mathbf{x} - \boldsymbol{\mu}))$$

where $\boldsymbol{\Sigma} \in \mathbb{R}^{D \times D}$ is a symmetric positive definite (SPD) matrix, $\boldsymbol{\mu} \in \mathbb{R}^D$ is the mean vector, $g : \mathbb{R}_+ \to \mathbb{R}$ is a function such that $\int_0^\infty t^{D/2-1} g(t) dt < \infty$ and

$$c_n = \frac{\Gamma(D/2)}{\pi^{D/2} \int_0^\infty t^{D/2-1} g(t) dt}.$$

where $\Gamma$ is the Gamma function (*i.e.* an extension of the factorial function). Gaussian Scale Mixtures (GSMs) distributions, which are known to describe the empirical distribution of wavelet coefficients of natural images [57], are a specific case of elliptical distributions [18], including the Gaussian distribution with $g(t) = \exp(-t/2)$, and the Student-t distribution with $g(t) = (1 + t/\kappa)^{-(\kappa+D)/2}$ and $\kappa \in \mathbb{R}_+$. Now, we define the empirical statistics of $N$ data samples $\mathbf{X} \in \mathbb{R}^{D \times N}$. We estimate the mean vector and the SPD matrix by standard empirical estimators, respectively

$$M_N(\mathbf{X}) = \frac{\mathbf{X}\mathbb{1}_N}{N} \quad \text{and} \quad C_N(\mathbf{X}) = G_N(\mathbf{X} - M_N(\mathbf{X})) \quad \text{with} \quad G_N(\mathbf{X}) = \frac{\mathbf{X}\mathbf{X}^{\mathrm{T}}}{N-1}. \quad (1)$$

where $\mathbb{1}_N = (1, \ldots, 1)^T \in \mathbb{R}^N$. Thanks to [19] Theorem 3, the function $g$ can be estimated by evaluating the distribution of the norm of the columns of $\bar{\mathbf{X}} = C(\mathbf{X})^{-1/2}(\mathbf{X} - M(\mathbf{X}))$. The density $h$ of these norms verifies

$$h(r) \propto r^{D-1} g(r^2).$$

We will later use this property to measure the empirical convergence of the functional parameter $g$.

**Wasserstein distance** The 2-Wasserstein distance (which we term Wasserstein hereafter) between two probability distributions $\mathbb{P}_X$ and $\mathbb{P}_Y$ can be defined (Remark 2.14 [38]) as

$$W_2(\mathbb{P}_X, \mathbb{P}_Y)^2 = \inf_{X \sim \mathbb{P}_X, Y \sim \mathbb{P}_Y} \mathbb{E}\left(\|X - Y\|^2\right). \tag{2}$$

When the two probability distributions $(P_X, P_Y)$ are elliptical with the same $g$ function and their SPD parameters is equal to their covariance matrices, the Wasserstein distance depends only on their means and covariances

$$W_2(\mathbb{P}_X, \mathbb{P}_Y)^2 = \|\boldsymbol{\mu}_X - \boldsymbol{\mu}_Y\|^2 + \mathcal{B}(\boldsymbol{\Sigma}_X, \boldsymbol{\Sigma}_Y)^2$$

where $\mathcal{B}(\boldsymbol{\Sigma}_X, \boldsymbol{\Sigma}_Y)^2 = \text{Tr}(\boldsymbol{\Sigma}_X + \boldsymbol{\Sigma}_Y - 2(\boldsymbol{\Sigma}_X^{1/2}\boldsymbol{\Sigma}_Y\boldsymbol{\Sigma}_X^{1/2})^{1/2})$ is the Bures metric between $\boldsymbol{\Sigma}_X$ and $\boldsymbol{\Sigma}_Y$ [38]. To give some intuition, it is worth noting that when $\boldsymbol{\Sigma}_X$ and $\boldsymbol{\Sigma}_Y$ commute (*i.e.* $\boldsymbol{\Sigma}_X\boldsymbol{\Sigma}_Y = \boldsymbol{\Sigma}_Y\boldsymbol{\Sigma}_X$),

$$\mathcal{B}(\boldsymbol{\Sigma}_X, \boldsymbol{\Sigma}_Y) = \left\|\boldsymbol{\Sigma}_X^{1/2} - \boldsymbol{\Sigma}_Y^{1/2}\right\|_f,$$

where $\|\ \|_f$ is the Frobenius norm between matrices. In 1D, it corresponds to the absolute value of the difference between the standard deviations.

**Texture statistics and synthesis algorithm** Except for a small subset of textures often called micro-textures [15], natural textures have non-Gaussian statistics. As for natural images, the statistics of their coefficients in a wavelet domain are well modeled by Gaussian Scale Mixtures (GSMs) [57]. Recent work suggests this may also be true when considering their coefficients at the different layers of a CNN [46, 17]. As mentioned above, GSMs are a specific case of elliptical distributions. Therefore, for a given texture, CNN activations at each layer $l$ can be represented by a triplet $(\boldsymbol{\mu}_l, \boldsymbol{\Sigma}_l, g_l)$ which defines the corresponding elliptical distribution. However, this is not sufficient to define a complete generative model of textures, because the triplet characterizes only the marginal distribution and not the joint distribution (*i.e.* spatial dependencies). For this reason, CNN-based texture synthesis methods [16], which consists of imposing target statistics to the feature vectors of a white noise image, exploit the spatial dependencies encoded implicitly in the neural network.

Here we adopt that approach, using the summary statistics of the elliptical distributions. Given the neural network (denoted by $\mathcal{F}$) and the statistics $(\boldsymbol{\mu}_l, \boldsymbol{\Sigma}_l, g_l)_l$ of a texture example $u$, we achieve synthesis by imposing the statistics to the feature vectors of an input white noise image $v$. The feature vectors of $v$ at layer $l$ are denoted by $\mathbf{X}_l^v = \mathcal{F}_l(v)$. We considered two different loss functions that when minimized as a function of an input noise $v$ will generate a new texture example. The first loss function, called "Gram" (previously used by Gatys *et al.* [16]), aims at enforcing the Gram matrix of the samples by minimizing the Euclidean norm of the difference with the target Gram matrix

$$L_{\text{G}}(u, v) = \sum_{l=1}^{L} \|G_{N_l}(\mathbf{X}_l^v) - \mathbf{G}_l^u\|_f^2 \tag{3}$$

where $\mathbf{G}_l^u = \boldsymbol{\Sigma}_l^u + \boldsymbol{\mu}_l^u \boldsymbol{\mu}_l^{u\,\text{T}}$ and $G_{N_l}$ is defined in Equation (1). The second is the "Wasser" loss and aims at minimizing the Wasserstein distance between the input and the target feature vectors

$$L_{\text{W}}(u, v) = \sum_{l=1}^{L} \|M_{N_l}(\mathbf{X}_l^v) - \boldsymbol{\mu}_l^u\|^2 + \mathcal{B}(C_{N_l}(\mathbf{X}_l^v), \boldsymbol{\Sigma}_l^u) \tag{4}$$

where $M_{N_l}$ and $C_{N_l}$ are defined in Equation (1). The Wasserstein distance is only an approximation here because there is no reason why the statistics of the input and the target feature vectors have the same $g_l$ functions at each layer $l$. Yet, we find in Section 3 that this is empirically enough to match the distribution even without any constraint on $g_l$. Using these loss functions the synthesis of a texture $u$ is achieved by estimating

$$\bar{v}_{\text{m}} = \text{argmin}_v \, L_{\text{m}}(u, v)$$

for $\text{m} \in \{\text{G}, \text{W}\}$. Importantly, this problem can be solved by gradient descent (backpropagation) initialized from a white noise image $\bar{v}_0$.

**Texture interpolation** The Wasserstein distance makes the set of probability measures a metric space (Proposition 2.3 [38]). Therefore, it offers a proper framework to perform texture interpolation because it allows to define the barycenter of $K$ probability measures $(\mathbb{P}_{X_k})_k$ with weights $(\lambda_k)$ by

$$\bar{\mathbb{P}} = \text{argmin}_{\mathbb{P}} \sum_{k=1}^{K} \lambda_i W_2(\mathbb{P}, \mathbb{P}_{X_k})^2 \quad \text{where} \quad \sum_{i=1}^{K} \lambda_i = 1.$$

Note that there is an alternative method when using the Wasserstein distance, see supplementary Section 2. Unlike $L_\mathrm{W}$, the Gram loss function $L_\mathrm{G}$ is not derived from a proper metric over the space of probability distributions (*e.g.* Gaussians distributions parametrized by $(\Sigma, \mu)$ and $(\Sigma - \mu\mu^\mathrm{T}, 2\mu)$ are different while being associated to the same Gram matrix). However, we can apply a similar heuristic to define interpolation between multiple textures $(u_k)_k$ weighted by $(\lambda_k)$

$$\bar{v}_\mathrm{m} = \operatorname*{argmin}_v \sum_{k=1}^K \lambda_k L_\mathrm{m}(u_k, v) \quad \text{where} \quad \sum_{i=1}^K \lambda_i = 1$$

and for $\mathrm{m} \in \{\mathrm{G}, \mathrm{W}\}$. This problem is also solved by gradient descent (backpropagation) initialized from a white noise image $\bar{v}_0$. The use of different loss functions defines different geometries over the space of probability distributions (even if it does not make it a proper metric space).

As a consequence, these loss functions will lead to the synthesis of different texture mixtures because the barycenters will lie on different geodesics. In practice, we interpolate between $K = 2$ textures, then $\lambda_1 = t$ and $\lambda_2 = 1 - t$ with $t \in [0, 1]$.

## 3 Results

First, we present our results on natural texture statistics and the Wasserstein framework. Then, we compare qualitatively our interpolation results to the PS algorithm [39], as well as different CNN architectures and the previously used Gram loss. Finally, we demonstrate with psychophysics and neurophysiology experiments how our texture interpolation algorithm can be used to probe biological vision. We used 32 natural textures from the dataset of [7] and 32 natural images from BSD [1].

**Statistics of natural textures** Figure 2, top, shows a successful example of texture synthesis by matching the mean and covariance matrix of the CNN activations (eq. (4)). In the Wasserstein framework, if the CNN activations of the example texture are well described by elliptical distributions then matching their mean vectors and covariance matrices is sufficient to match the full distributions. Using the method described in supplementary Section 1, we found that textures are more elliptically distributed than natural images. Specifically, first, the distribution $\nu_\mathrm{tex}$ of CNN activations (Figure 2, third row, orange) was as concentrated as that of wavelet coefficients (bottom row, orange) which are known to be approximately elliptically distributed (*i.e.* as GSMs [57, 8, 47]), although both were less concentrated than Gaussian random vectors (green). Second, the activations of natural textures were more concentrated than those of natural images (Figure 2, blue histograms), and synthesis failed for the example image (Figure 2, top). Similar to wavelets [8, 47], CNN activations of natural images may also be better described by

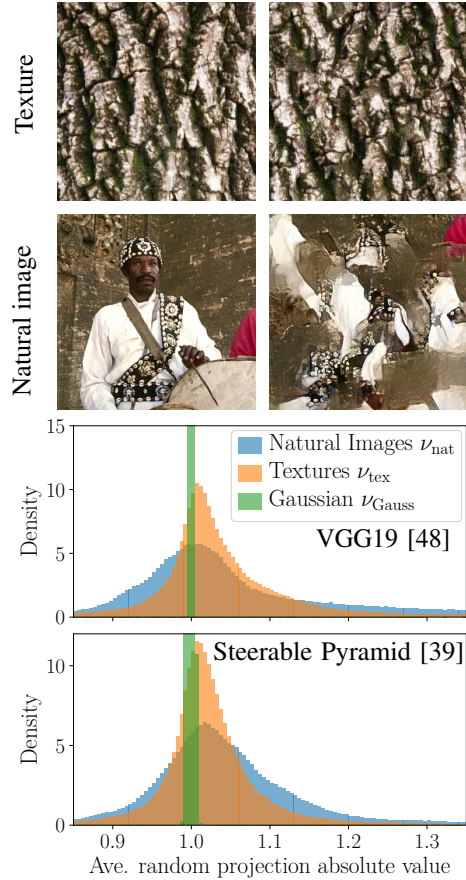

*Figure 2: Top two rows: examples of synthesis using our method (eq. (4)). Bottom two rows: quantification of the elliptical symmetry of natural images' and textures' CNN/wavelet activations at a single layer/scale. The narrower the more elliptical. Differences in the histograms for Gaussians are due to the different dimension $D$.*

mixture distributions with some components corresponding to textures [55]. We suggest that the higher ellipticity of CNN activations of natural textures compared to those of natural images can explain the success of deep texture synthesis methods like [16].

**Empirical convergence of the radial function** The Wasserstein loss function (eq. (4)) corresponds to the Wasserstein distance (eq. (2)) only when both distributions are from the same elliptical family *i.e.* when they have the same norm density $h$. In practice this is not the case because we initialize our optimization with a white noise texture, for which CNN activations are from a different elliptical

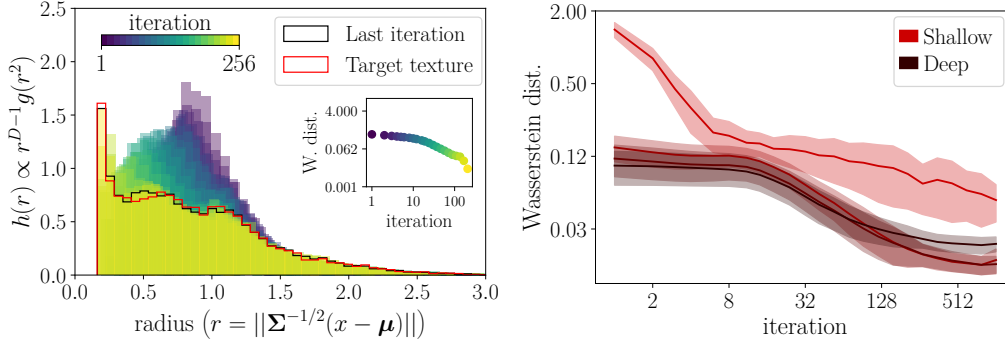

*Figure 3: Empirical convergence of the radial distribution $h$ (and therefore $g$). Left: example for a single texture at a single layer. The histograms represent the distribution of $h$ at each iteration (color coded from blue to yellow). The inset shows the 1D Wasserstein distance between the histogram at current iteration and the histogram corresponding to the target texture. Right: average 1D Wasserstein distance, across 32 textures at different layers. Shaded areas: 95% c.i.*

family. Yet, we find empirically that during training the distribution $h$ converges towards that of the target texture. This is illustrated for one example texture in Figure 3, left, and quantified across different textures and CNN layers in Figure 3, right. This result also holds when the network has random weights (not shown), suggesting that the network architecture and the mean and covariance of its activations encode the information corresponding to distribution $h$.

**Comparison of interpolation methods** The Wasserstein loss (eq. (4)) offers a natural way to perform texture interpolation using the geodesics defined by the corresponding Wasserstein metric (Section 2, §6). We compare the interpolation obtained with the Wasserstein loss using VGG19 with trained and untrained weights, and a single layer multiscale architecture [52]. We also compare to the interpolation obtained with the Gram loss using VGG19 trained weights [16] and to a variant of the PS algorithm [39] (extension to interpolate color textures [54]). Figure 4 illustrates our main qualitative observation (see also supplementary Figures 7, 8): both the Wasserstein loss interpolation and the PS algorithm generate a perceptually homogeneous mixture of the target textures, whereas the Gram loss interpolation generates textures that are composed of discrete patches of the target textures. Interestingly, both architectures with untrained weights generate more patchy textures, similar to the Gram loss.

**Paths of texture interpolation** We argue that the interpolation results in Figure 4 for the PS algorithm and the Wasserstein loss with trained weights are more perceptually meaningful, because the interpolated textures preserve stationarity, complying with the hypothesis that texture perception is statistical. As our goal is to study visual perception, we compared the paths of interpolation of the Wasserstein loss and the PS algorithm (which has been used in previous experimental studies [14,

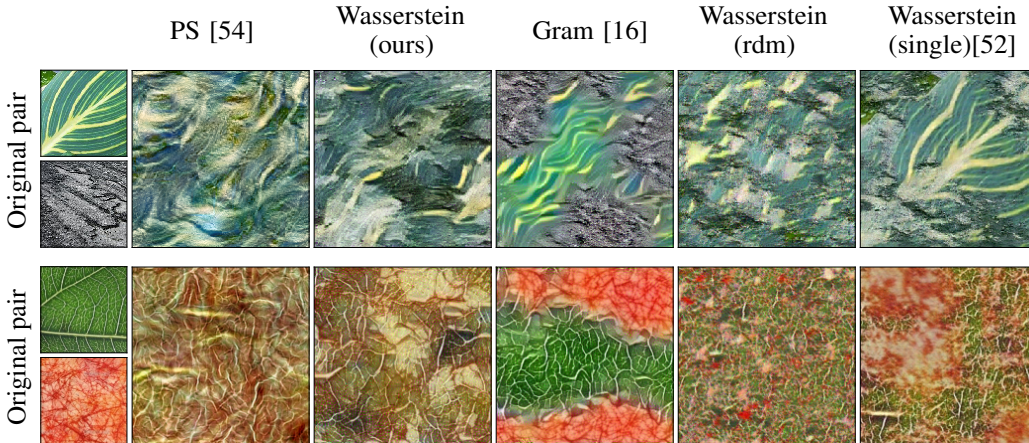

*Figure 4: Comparison of textures interpolation methods. PS: the PS algorithm [39] extended to color texture interpolation [54]. Wasserstein (ours), (rdm) and (single): our method eq. (4), using respectively VGG19 pretrained, VGG19 with random weights, and a single layer multiscale architecture [52]. Gram: as in [16] eq. (3). Interpolation weight $t = 0.5$.*

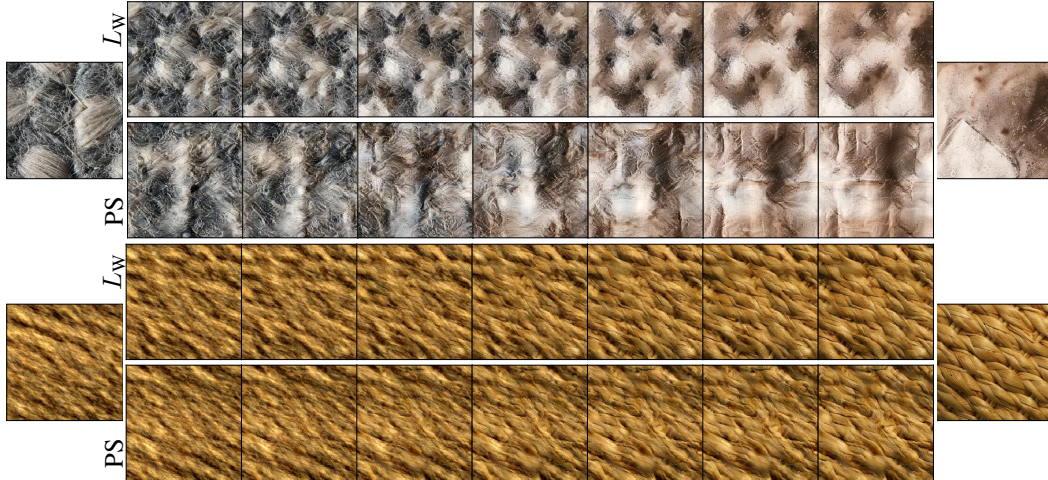

*Figure 5: Examples of texture interpolation paths for the Wasserstein loss ($L_W$) using VGG19 with trained weights, and the PS algorithm extended to interpolation of color textures [54]*

35]). Interpolation results are shown in Figure 5 (and also in supplementary Figures 9–12). Overall, these interpolations are perceptually smooth for both algorithms, *i.e.* textures synthesized with close weights appear almost indistinguishable. The interpolations are also quite similar despite the edge artifacts present in PS results (which are due to the periodic boundary implicitly assumed by the Fourier transform used in the complex steerable pyramid [39]). In addition to interpolating between different textures, Figure 5 shows an example of interpolation between a natural texture and a spectrally matched Gaussian texture. This could be useful for vision research studies, because neurons in V1 are thought to be mainly sensitive to spectral cues, whereas higher areas V2/V4 are sensitive to higher-order statistics of these spectral cues [14, 35]. Our qualitative comparison suggests that both our method and PS interpolation may be suitable to probe vision (no patches, smooth interpolation). Yet, our method has the advantage that it relies on simple statistics (mean and covariance) of CNN activations combined with OT, and thus offers a well-grounded mathematical framework for further modeling of perception and neural activity.

**Perception of interpolated textures** To validate the intuition that our interpolation produces perceptually meaningful results, we measured the perceptual scale associated to the interpolation weight using the MLDS protocol [32, 29].

*Protocol* The experiment consists of 2-alternate forced choice trials. Participants are presented with 3 stimuli with parameters $t_1 < t_2 < t_3$ and are required to choose which of the two pairs with parameters $(t_1, t_2)$ and $(t_2, t_3)$ is the most similar. We used two sets of three textures (see supplementary Figure 6): (i) a first set where we interpolate between the stationary Gaussian synthesis ($t = 0$) and the naturalistic texture ($t = 1$, as in Figure 5 bottom); (ii) a second set where we interpolate between two arbitrary textures (as in Figure 5 top). All stimuli had an average luminance of 128 (range $[0, 255]$) and an RMS contrast of 39.7. For each texture pair, we use 11 equally spaced ($\delta_t = 0.1$) interpolation weights. To ensure that stimulus comparisons are above the discrimination threshold we only use triplets such that $|t_i - t_j| \geqslant 2\delta_t$. For each texture set (i) and (ii), a groups of 8 naive participants performed the experiment. Participants were recruited through the platform prolific[2] and performed the experiments online. The protocol was approved by the Internal Review Board of the Albert Einstein College of Medicine. Monitor gamma was corrected to 1 assuming the standard value of 2.2.

*MLDS model* The MLDS model assumes that the observer uses a perceptual scale that is an increasing function $t \mapsto f(t) \in [0, 1]$ to perform the task. At each trial, the observer makes a stochastic judgement whether or not $f(t_1) - 2f(t_2) + f(t_3) + \varepsilon < 0$, where $\varepsilon$ is the observer noise modeled as a zero-mean Gaussian variable with standard deviation $\sigma > 0$.

*Results* For each texture, we fitted the MLDS model on participants' data pooled together using the standard method [29] (Figure 6). We also fitted the model to individual participant data (supplementary Figure 4 and 5). First, we found that the measured perceptual scale is meaningful, because confidence intervals for human participants (Figure 6, colored shaded areas) are tight compared to

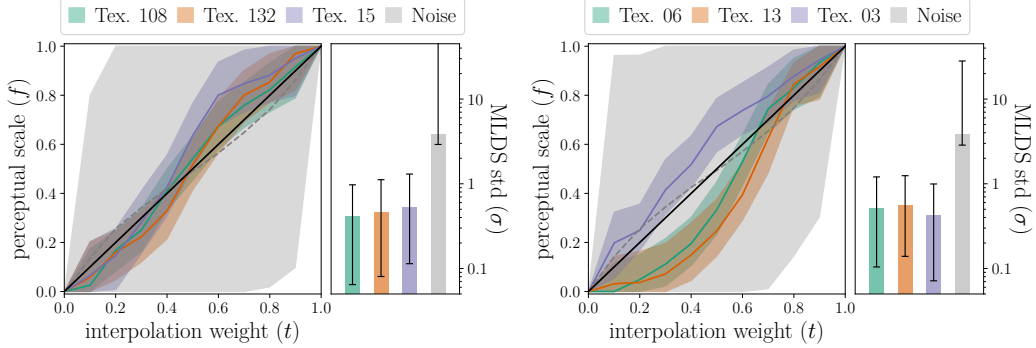

*Figure 6: Perceptual scale of the interpolation weights and the corresponding observer model standard deviation. Error bars are 99.5% bootstrapped c.i.. Left: Stationary Gaussian to naturalistic interpolation. Right: Interpolation between arbitrary texture pairs.*

an observer that would answer randomly (gray shade) and despite the additional inter-participant variability. This is also confirmed by the estimated standard deviations of the MLDS model (Figure 6, bar plots) that are one order of magnitude lower for the participants than for the random observer. For the first set of textures (i), perceptual scales have roughly an S-shape. Individual fits (supplementary Figure 4)) reveal that there are in fact 3 underlying behaviors: (a) linear for participants 1, 2 and 4; (b) S-shape for participants 3,7 and 8 and (c) top-asymmetric concave for participants 5 an 6. In the second set of textures (ii), perceptual scales have two different behaviors: (d) symmetric concave for texture pair 03 and (e) bottom-asymmetric convexe for texture pairs 06 and 13. Individual fits (supplementary Figure 5)) could be characterized by these behaviors. Taken together these results show an insight of the diversity of behaviors. More participants and textures will be required for a complete characterization and to further understand the implications for perception.

**Neural coding of interpolated textures** To illustrate how our interpolation method can be used to probe neural coding, we analyzed the spiking activity of simultaneously recorded V1 (N=6) and V4 (N=5) neurons in macaque monkeys, in response to 3 distinct sets of interpolated textures.

*Protocol* Textures were interpolated between synthesized naturalistic textures ($t = 1$) and their spectrally matched Gaussian counterpart ($t = 0$) at 5 different weights ($t = 0.0, 0.3, 0.5, 0.7, 1.0$). All stimuli had their luminance normalized as in the MLDS experiment and were presented at 5 different sizes ($2°$, $4°$, $6°$, $8°$, $10°$) on a CRT monitor. Recordings were conducted in an awake, fixating adult male macaque monkey (*Macaca fascicularis*) implanted with "Utah" arrays in V1 and V4 [27]. A successful trial consisted of the subject maintaining fixation over a central $1.4°$ window for 1.3 seconds. During this time we presented a sequence of 3 textures displayed for 300-ms each and immediately followed by a 100-ms blank screen.

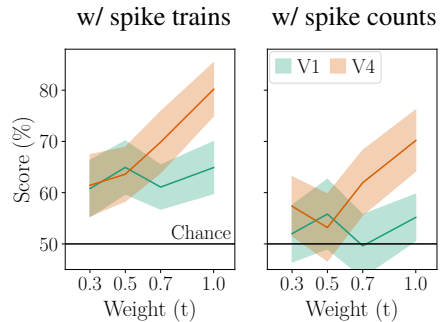

*Figure 7: Decoding of weights $0$ and $t = 0.3, 0.5, 0.7$ and $1.0$, averaged over 3 textures, from V1 and V4 neural activity. Error bars are 95% c.i.. Left: using full spike trains+smoothing+PCA. Right: using raw spike counts.*

*Data analysis* We decoded the interpolation weight from both spike trains and spike counts by Linear Discriminant Analysis using scikit-learn [37]. Specifically, we considered three pairs comprising one texture and the corresponding spectrally matched Gaussian texture, and we decoded the following pairwise weight conditions $(t_1, t_2) = (0.0, 0.3), (0.0, 0.5), (0.0, 0.7), (0.0, 1.0)$ for each texture pair. For spike trains, we first reduced the dimensionality using Principal Component Analysis, and chose the number of components that maximized the averaged cross-validated (5 folds) classification performance.

*Results* We asked if neurons are sensitive to the interpolation parameter, and if sensitivity is different across areas. The results in Figure 7 suggest that adding naturalistic content to a stationary Gaussian texture, while preserving its power spectrum, does not increase the stimulus-related information in V1, whereas it increases approximately linearly with the weight in V4. This is true both when using the spike count during the stimulus presentation (Figure 7 left) or the full spike train (Figure 7 right).

This decoding performance corroborate the fact that neurons are tuned to the interpolation weight in V4 and not in V1 (see supplementary Figure 3). The decoding performance is overall higher, and the trend more linear for V4, when using the full spike train which indicates that relevant information is encoded dynamically. These results thus offer a basis to further link the measured perceptual scale to neural activity in the visual cortex.

# 4 Discussion

We studied texture statistics across layers of CNNs, and found that they are more elliptical than natural images. Based on that finding, we showed that texture synthesis can be cast as a minimization of the Wasserstein distance to the distribution of CNN activations of a target texture. The proposed framework offers a geometric interpretation of the space of textures, and affords a precise definition of distance and interpolation between textures based on optimal transport. In particular, it allows one to define a neighborhood ($\varepsilon$-environment) that is compatible with simple summary statistics (see [52]). Our empirical analysis of synthesis and interpolation suggests that CNN weights and architecture both have an effect on the interpolation paths of textures. When the CNN is trained the paths are perceptually smoother and consist of stationary homogeneous textures. This might be because a trained network provides a smoother approximation of the manifold of textures. To our knowledge, there is no comparison of the Gram *vs* Wasserstein loss for texture synthesis. The reason is that, differences are not visible at first sight on the synthesized textures without sampling interpolation of textures. Such differences may be crucial for visual perception studies but less for computer graphics. We also found that the linear interpolation in the inhomogeneous space of PS summary statistics generates homogeneous textures. However, the combination of CNN and optimal transport has the practical advantage of a homogeneous feature space and simpler distributions, thus offering a well-grounded mathematical framework for characterizing and modeling biological visual processing. Yet, our work is limited to textures while a full understanding of natural image space is crucial to further understand visual perception [23, 10]. Also, we didn't explore the possibility of improving the network architecture to produce similar quality textures using less parameters like feature variances and means or feature means only [12, 11]. We demonstrated the applicability with perceptual and neurophysiology experiments, and found preliminary evidence that our interpolation based on probability distributions influence both perception and neural activity. Previous work [14, 58] focused on the perceptual sensitivity of "naturalness" (when the weight $t$ goes from 0 to 1) and showed that it is partly predicted by neuronal responses in V2. In comparison, the MLDS protocol will allow for the measurement of a full perceptual scale, not just perceptual sensitivity. We expect to further relate the perceptual scale to recordings in the visual cortex. In addition, we do not limit our study to the perception of "naturalness" and we propose to measure the perception of interpolation path between arbitrary texture pairs. Such interpolation paths should provide more fine-grained information about the perception of the texture space geometry than previous massive data collection [35].

## Broader Impact

Any protocol that quantifies perception could potentially be turned into a diagnostic tool. Apart from that, our work does not present any foreseeable societal consequence.

## Acknowledgements

We thank Anna Jasper for help with neuronal recordings and Pascal Mamassian for suggesting the use of MLDS and for fruitful discussions. JV and RCC are supported by NIH (NIH/CRCNS EY031166).

## Footnotes

[1]https://github.com/JonathanVacher/texture-interpolation

[2]https://www.prolific.co

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
