[Supplementary Material]

# Texture Interpolation for Probing Visual Perception

**Jonathan Vacher**[*]
Albert Einstein College of Medicine
Dept. of Systems and Comp. Biology
10461 Bronx, NY, USA
jonathan.vacher@ens.fr

**Aida Davila**
Albert Einstein College of Medicine
Dominick P. Purpura Dept. of Neuroscience
10461 Bronx, NY, USA
adavila@mail.einstein.yu.edu

**Adam Kohn     Ruben Coen-Cagli**
Albert Einstein College of Medicine
Dept. of Systems and Comp. Biology, and
Dominick P. Purpura Dept. of Neuroscience
10461 Bronx, NY, USA
adam.kohn@einsteinmed.org
ruben.coen-cagli@einsteinmed.org

## Abstract

Supplementary material for the paper "Texture Interpolation for Probing Visual Perception".

## Contents

## 1  Measuring ellipticity of empirical distributions

Despite existing tests for ellipticity [1, 2, 4], there is no simple and standard quantification of ellipticity for high-dimensional empirical distributions. Given the whitened samples $\bar{\mathbf{X}}$ of $\mathbf{X}$, measuring the ellipticity of the distribution of $\mathbf{X}$ comes down to measuring the sphericity of the distribution of $\bar{\mathbf{X}}$. This can be achieved by quantifying the uniformity of the distribution of $\bar{\mathbf{U}} = (\bar{X}_1/\|\bar{X}_1\|, \ldots, \bar{X}_N/\|\bar{X}_N\|)$ over the unit $D$-sphere and its independence from $(\|\bar{X}_1\|, \ldots, \|\bar{X}_N\|)$. We will limit ourselves to measure the uniformity of the distribution of $\bar{\mathbf{U}}$. To achieve this, we consider $\tilde{\mathbf{X}} = (R_1\bar{U}_1, \ldots, R_N\bar{U}_N)$ where $R_i$ are drawn from a $\chi$ distribution with $D$ degrees of freedom. Thus, if the distribution of $\bar{\mathbf{U}}$ is uniform on the unit $D$-sphere, the distribution of $\tilde{\mathbf{X}}$ is Gaussian. From there, we consider the distributions $(\nu_i)_i$ of the absolute value of the projection $(|\langle \tilde{X}_1, \bar{U}_i \rangle|, \ldots, |\langle \tilde{X}_N, \bar{U}_i \rangle|)_i$. When the distribution of $\tilde{\mathbf{X}}$ is Gaussian, all the distributions $\nu_i$ are histograms of the same half-normal distribution. Therefore, the distribution $\nu$ of their parameter estimates is concentrated around 1 with high

---

[*]JV is now based at Laboratoire des Systèmes Perceptifs (LSP), Département d'études cognitives, École Normale Supérieure, PSL University, 75005 Paris, France

precision (see main text Figure 2). We will use this method to quantify ellipticity of CNN/wavelet activations of natural images and textures, using the heuristic that the more concentrated the distribution $\nu$, the more elliptically distributed the samples $\mathbf{X}$.

Figure 1: Scatter plots of random sample from a Gaussian distribution (blue colored dots on the left frame) and from a mixture of Gaussian distributions (blue and orange dots on the right frame). The four pairs of scatter plot corresponds to different dimensions as indicated. In dimension higher than 3, a pair of axis is chosen randomly.

Figure 2: Distribution $\nu$ computed using the method described in Section 1. The four pairs of histogram correspond to different dimensions as indicated. The blue histogram is for the Gaussian samples while the orange histogram is for Gaussian mixture samples.

We tested our method on simulated samples of a Gaussian (elliptic) and a Gaussian mixture distribution (non-elliptic) for different dimensions. In low dimension, the scatter plots (Figure 1 top) show how Gaussian mixture is non-elliptic by depicting the superposition of two elliptic dot clouds. In high dimension, the scatter plots (Figure 1 bottom) of two random axis is much less explicit about non ellipticity. This is because most pairs of axis have similar variances. In contrast, the histograms (Figure 2) of $\nu$ verifies our heuristic: the distribution $\nu$ is more concentrated for elliptically distributed samples than for non-elliptically distributed samples.

## 2 Alternative Wasserstein Interpolation

For the Optimal Transport framework, an alternative method for interpolating between two textures is possible. This is because the mean and covariance of the interpolating texture can be written in closed form

$$\mu_{tX+(1-t)Y} = t\mu_X + (1-t)\mu_Y \quad \text{and} \quad \Sigma_{tX+(1-t)Y} = t\Sigma_X^{1/2} + (1-t)\Sigma_X^{-1/2}\left(\Sigma_X^{1/2}\Sigma_Y\Sigma_X^{1/2}\right)^{1/2}$$

where $((\mu_X, \Sigma_X), (\mu_Y, \Sigma_Y))$ are the means and covariances of the interpolated texture features. We did not fully explore this approach because we wanted to compare to other methods where this approach is not feasible. The generated textures were in general noisier than the weighted loss approach. We hypothesized that this is due to the multiple necessary regularization used to compute $\Sigma_{tX+(1-t)Y}$.

## 3 Neurophysiology

The reason why we have increasing decoding performances with spike counts in V4 and not in V1 is because the recorded neurons are tuned to higher order statistics in V4 and not in V1. Except for texture 15, the spike count of all neurons increases as the texture has more higher-order statistical content while it remains approximately flat in V1 for $t \geqslant 0.3$.

Figure 3: Tuning curves to interpolation weights of five recorded neurons of V1 (top) and V4 (bottom).

# 4 Individual Psychometric Results

Figure 4: Stationary Gaussian to naturalistic interpolation. Perceptual scale of the interpolation weights and the corresponding observer model standard deviation for the 8 participants. Error bars are 99.5% bootstrapped c.i..

Figure 5: Interpolation between arbitrary texture pairs. Perceptual scale of the interpolation weights and the corresponding observer model standard deviation for the 8 participants. Error bars are 99.5% bootstrapped c.i..

# 5 Stimuli

Figure 6: Stimuli used in our experiments. From top to bottom: Tex 15, Tex 108, Tex 132, Tex 03, Tex 06 and Tex 13.

# 6 Supplementary Synthesis Results

Figure 7: Comparison of textures interpolation methods. PS: the PS algorithm [5] extended to color textures. $L_W$, $L_W$ (rdm), and $L_W$ (single): our method, using respectively VGG19 pretrained, VGG19 with random weights, and a single layer multiscale architecture [6]. $L_G$: as in [3]. Interpolation weight $t = 0.5$.

Figure 8: Comparison of textures interpolation methods. PS: the PS algorithm [5] extended to color textures. $L_W$, $L_W$ (rdm), and $L_W$ (single): our method, using respectively VGG19 pretrained, VGG19 with random weights, and a single layer multiscale architecture [6]. $L_G$: as in [3]. Interpolation weight $t = 0.5$.

Figure 9: Examples of texture interpolation paths for the Wasserstein loss using VGG19 with trained weights.

Figure 10: Examples of texture interpolation paths for the Wasserstein loss using VGG19 with trained
weights.

Figure 11: Examples of texture interpolation paths for the PS algorithm.

Figure 12: Examples of texture interpolation paths for the PS algorithm.

Figure 13: Examples of the sample diversity of texture interpolation paths for the Wasserstein loss using VGG19 with trained weights.

Figure 14: Examples of the sample diversity of texture interpolation paths for the Wasserstein loss using VGG19 with trained weights.

Figure 15: Examples of the sample diversity of texture interpolation paths for the Wasserstein loss using VGG19 with trained weights.

Figure 16: Examples of the sample diversity of texture interpolation paths for the Wasserstein loss using VGG19 with trained weights.