[Reviews · NeurIPS 2020]

Review 1

Summary and Contributions: This paper proposes a new way of characterizing textures via elliptical distributions and leverages on Optimal Transport Theory to interpolate along the representational geodesic of an encoded VGG19 between any two textures. Authors not only theoretically motivate their approach, but also empirically verify it with human psychophysics and monkey neurophysiology. -- Post Rebuttal --: I've kept my score and am happy with the rebuttal sent by the authors addressing some of my minor concerns.

Strengths: *1) This last empirical validation (have human psychophysics and monkey neurophysiology) is critical and differentiates this paper over the myriad of works on texture synthesis (and potentially style-transfer) given that there are very little rigorous and/or highly controlled empirical tests of synthesis procedures and their links to perception. This paper addresses these by actually running the 2 experiments previously mentioned and comparing their results to the PS model, and showing other results of qualitative interpolations for different variations of their model (Lw) *2) The interpolation method is theoretically motivated *3) The authors make an excellent job motivating the need for research on this area to be done (linking texture synthesis to perception, beyond 'eye-balling results' or AMT studies as is generally done in Computer Vision). In addition they do a great job reviewing the cross-disciplinary literature given the nature of the problem they are working on (although there are a couple of references missing -- see Relation to Prior Work section). *4) The paper is interdisciplinary by nature and puts back the "Neural" part in Neural Information Processing Systems (NeurIPS). It is a mix of engineering, machine learning, perception and neuroscience -- and although I have some minor reservations explained through out my review I am still inclined for this particular reason to give it a high score.

Weaknesses: * See section (Clarity). * 2 observers are used in human psychophysical task and the number of trials is not stated. Also not stated if observers were authors or naive to goals/outcome of experiment (unless this is mentioned in the Supplement and I might have missed this). * For better or for worse, there is a lot going on -- and it feel like the pages 5 and 8 were crammed up. I wonder if maybe the theory of OT (pages 3-4)can be reduced to 1 page, so that there can be a more thorough discussion, and explanation of the results and metrics rather than having the reader refer to the Supplementary Material consistently for some minor details.

Correctness: Why are the neurophysiological experiments done on V1 vs V4 instead of V1 vs V2 -- as it is known that V2 encodes information in a way that is similar to texture (Ziemba et al. Selectivity & Tolerance [...]). Or is the goal here to make a dissociation between the firing of neurons before (V1) and after (V4) sending/receiving inputs to V2?

Clarity: * Yes -- although pages 3-4 are non-trivial to follow, I wonder, given the interdisciplinary nature of the paper -- will this paper still be accessible to less theoretical audiences in neuroscience and perception (ex those who read / submit to: Journal of Vision, CCN, VSS)? * Figures 1 and 2. Hmm, are these figures "key" to the main body of the paper? It seems like they are somewhat supplementary, or should not be placed as "Figure 1" (usually the most important figure of the paper that potentially elaborates on the approach and/or methods given the narrative). Maybe another key figure is missing and Figures 1/2 should actually be displaced to Figures 2/3 etc [...] *Minor comment: Figure 3 should be more clear and direct in specifying that L_w is the proposed method along with it's variations. Something like L_w (ours/proposed). * I am not sure what I should be looking for in Figure 5. It seems like lower is better for the colored bars, but there is little explanation of what the perceptual scale $(f)$ means besides a couple of citations/references.

Relation to Prior Work: Yes, although there are some missing references in the Introduction / Section 1: -- Texture Manifolds (ICML 2017) Bergmann et al. : Texture Interpolation through a spatially period GAN across a family of several textures although no links are done to neurophysiology. -- Towards Metamerism via Foveated Style Transfer (ICLR 2019) Deza et al.: Image to Texture interpolation in the encoded space of a VGGNet for a log-polar receptive field with some links to neurophysiology, and perception via human psychophysics. -- Geodesics of Learned Representations (ICLR 2016) Henaff & Simoncelli : Image to Image interpolation via the representational geodesic and an enforced L2 loss in the pixel space. No links to neurophysiology. Other than adding those citations in the Introduction or Discussion, I am highly enthusiastic about accepting this paper due to its fresh perspective, interdisciplinary nature, and novelty of texture interpolations with empirical links to neurophysiology via macaque single-cell recordings and perception via human psychophysics.

Reproducibility: Yes

Additional Feedback: Would it be possible to show interpolation paths (Figure 4) for different starting noise seeds (of the synthesized image) to visualize the diversity in the interpolating paths? Typo[?]: Figure 5 (left) inset right shows P1,P2,P1,P3,P4,P4,N but should only be P1, P2 and N like it's right counterpart for arbitrary texture pairs. How do the results from Line 295-298 fit with the paper "Texture Synthesis Through Convolutional Neural Networks and Spectrum Constraints" (ICPR, 2016) by Liu, Gousseau & Xia. I also wish the discussion section would have been expanded, it seems like the paper (for better or for worse) tried to squish many interesting results in the main body (8 pages).


Review 2

Summary and Contributions: Proposes a geodesic distance between elliptically symmetric distributions over CNN filter outputs as a way to interpolate between textures. Evaluates results by psychophysical similarity and neural responses.

Strengths: An interesting method and nice results!

Weaknesses: Perhaps a bit too heavy handed and not very transparent computational approach. Although the textures look nice - it does the job - do you really think the brain models and represents textures this way? It seems to be there must be a simpler approach that could more efficiently capture the structure in these textures. (btw this isn't a strong weakness, just a personal reflection that makes me somewhat uncomfortable about this approach)

Correctness: as far as I can tell

Clarity: Yes.

Relation to Prior Work: Yes.

Reproducibility: Yes

Additional Feedback:


Review 3

Summary and Contributions: <<<Update after author response: In the author response, the authors suggest that testing additional models via psychophysics should be future work, such that this paper would be considered the technical reference. Given how much is already in the submission, I agree with this, although the limitations of the experiments should be made more clear in the final submission. The authors suggestion to expand on the technical details in the supplement will be helpful in understanding the presented method. One particular concern I had about technical details was also addressed in the rebuttal -- the authors clarify that the distributions of natural textures are only "more" elliptical than images, and that they are not actually elliptical. Although I don't think that this invalidates the presented results, as empirically the model generates realistic looking textures, this limitation should be acknowledged (ie lines 203-206 are somewhat misleading). Even with these limitations, I think that this is a good submission and an excellent fit for NeurIPS. >>> This paper uses optimal transport theory to develop a texture synthesis method that provides interpretable texture interpolations. The authors empirically present results suggesting that VGG19 activations to textures are elliptically distributed (while they are not for natural images). Using this, the authors propose a Wasserstein based distance measure to synthesize textures from the CNN activations and smoothly interpolate between multiple different textures. These results are compared to interpolations from other texture syntheses techniques and to the activations produced by a random network. The authors present a psychophysics study designed to measure if the interpolation weights for their presented model correspond to a perceptual similarity scale. The authors also present neural data to see if neurons are sensitive to the interpolation parameters.

Strengths: The work provides some theoretical basis for why texture synthesis from CNNs works as well as it does by claiming that this is because CNN activations for texture are elliptically distributed. This is of high significance to researchers studying texture perception. The study also ties together perceptual and neural experiments as validation of their results, providing diverse evidence that the presented texture interpolation can be used for studying visual responses (of interest for neuroscience studies). In the sense, the work is a good fit for the NeurIPS community. The authors also present the Wasserstein loss function for texture generation, and compare the interpolation methods to the Gram matrix method. I found the results of these two interpolation types particularly interesting, as it suggests that the LW loss might be better than the LG loss for texture synthesis. I am not currently aware of other papers directly comparing an Gram Matrix based loss with an Wasserstein distance based loss, making it significant and novel.

Weaknesses: The presented psychophysics study seems underpowered. The authors only include psychophysics and neural results for the LG model, so there is no direct comparison between interpolations of different texture models other than the presented demos. The authors argue that their method using the Wasserstein loss is better than previous interpolations methods, however their results are similar for the interpolations with PS statistics in Figure 3 and 4. They argue that fact that their statistics as "simpler" makes it a better model. Although the measured statistics classes are just the mean and covariance, the CNN based statistics are inherently complex because they have been measured from a set of optimized filters and are not necessarily grounded in human perception, so I am not convinced by this claim.

Correctness: In the abstract, the authors claim "The comparison to alternative interpolation methods suggests that ours matches more closely the geometry of texture perception" however alternative models are not explicitly compared in the paper via psychophysics (the authors however convincingly argue that their method has more mathematical grounding than the LG based approach). As expanded on in the clarity section, I found it somewhat difficult to evaluate whether the method for measuring the ellipticity of the distributions was correct.

Clarity: Overall the paper is well written, however some sections of the paper were a bit hard to follow. Particularly, the "Statistics of natural textures" section was difficult to interpret -- what is being averaged over for the random projections constructing the distribution? The text says that the distribution is constructed from the CNN activations -- were these the activations measured for one texture or over a large texture set? And are the random projections computed for all layers or a single layer of the network? I also found it difficult to follow the explanation of measuring the ellipticity of empirical distributions. I appreciate that the authors have developed a test ellipticity given that (in their words) "there is not a simple and standard quantification for ellipticity for high-dimensional empirical distributions". Perhaps including a schematic figure illustrating the metric would give intuition.

Relation to Prior Work: The paper adequately reviews previous work on visual texture models and ways of interpolating between textures. Textures are also relevant for studying the human auditory system (ie sounds such as rain, wind, and fire), and "texture morphs" were previously used with human psychophysics to study auditory processing (see McWalter & McDermott Current Biology 2018). Discussion of the similarity should be included, as the psychophysics approach is very similar. It seems like the work should include further discussion of how the Wasserstein based loss is better than the LG loss (as the LG loss has been wildly used for visual texture synthesis), and state whether there are previous papers that looked at this comparison (or lack of them if it is the case).

Reproducibility: Yes

Additional Feedback: * How many parameters are being matched for the Wasserstein framework compared with the Gram framework (and how does this compare with previous texture models)? Is it possible that the difference in the L_G vs. L_W textures is due to over parameterization in the L_G model? * Is there a guarantee that the generated textures will be diverse? See Loaiza-Ganem, Goa, & Cunningham ICLR (2017) for a discussion on how some previous texture models using the gram matrix can fail this test (which maybe could contribute to the non-stationarity of the interpolated textures). * As discussed in the clarity section, I am unconvinced that Figure 1 fully demonstrates that the CNN texture distributions are elliptically distributed (which seems crucial to the main claims of the paper). Explicitly quantifying the similarity between the texture distribution from VGG19 and the Steerable Pyramid model would improve these results. * The authors use the LW and PS models for the psychophysics and neural experiments, because they claim that those methods better preserve stationarity. Although it seems apparent from the presented images, a perceptual experiment quantifying this intuition that the LW loss is better than the LG loss for interpolation would be of interest. * As acknowledged in the paper, a larger sample size for the perceptual experiments presented in Figure 5 would improve the results. * A summary of the optimization success for the gradient descent based texture synthesis should be included. Is it possible that the difference between models in Figure 3 comes from a difference in the optimization landscape, such that some are not as closely matched to the desired statistics as others?


Review 4

Summary and Contributions: The authors propose a method for visual texture analysis/synthesis based on the statistical properties of the response of CNNs. They show that this texture characterization leads to meaningful texture interpolation in psychophysical terms. Moreover they show that different regions of the visual brain have different sensitivity along the interpolation lines in the texture space. # POST REBUTTAL After reading the reviews and author's reply, I keep my positive score: the authors have offered clarifications for the concerns and questions I had. I think is very interesting work and hope to see it in the conference. #

Strengths: The texture representation proposed by the authors is a bridge between classical techniques such as [Portilla&Simoncelli00] and more recent deep-learning techniques like [Gatys et al.16]. Particularly interesting is the psychophysical evaluation of the technique, and the experimental use of the textures in stimulating different brain regions. It is important to stress that these experimental validations are not usual in pure machine-learning proposals and they are highly valuable. This should grant acceptance since the development of this kind of controlled stimuli to study the properties of biological vision is important for the NeurIPS community for a better quantitative understanding of the behavior of the visual brain.

Weaknesses: It is not clear that interpolation in the proposed representation is better than previous interpolations (e.g. Portilla & Simoncelli). The psychophysical results show that the proposed interpolation is meaningful, but not that it is better than others. Qualitatively speaking, it is not obvious that less patchy interpolations are better. Similarly, the physiological experiments to decode the texture blends are only done with the proposed interpolation. How these physiological results relate to previous studies where texture discrimination was analyzed at different cortical levels? (e.g. Freeman & Simoncelli Nature Neuroscience 2011). A problem of Gatys et al.16 is that they do not justify the different weights given to the differences between the Gram matrices of the signal at the different layers of the network. In this work, the textures are characterized by the mean and the covariance of the elliptical PDFs followed by the signal (at the different layers). It is not clear if all the layers have the same relevance and why. Experiments should be explained better. For instance, Fig.6 in relation to Supplementary fig.1. Does Fig.6 correspond to the average of all neurons while curves in fig.s1 correspond to specific neurons?. What is the rationale to consider two sets of stimuli in the psychophysical experiments of fig.5?.

Correctness: The statistical model and the experimental validation are correct. However, additional experiments would make the results more solid: as stated above, the results in the work suggest that the proposed textures are meaningful in psychophysical and physiological terms, but not more meaningful than previous texture representations.

Clarity: The text describes the main ideas fairy well, nevertheless, more details would be required for reproducibility. The code they refer to may clarify this point.

Relation to Prior Work: Yes: the texture representation proposed by the authors is a bridge between classical techniques such as [Portilla&Simoncelli00] and more recent deep-learning techniques like [Gatys et al.16]. The part that is missing is the relation of the experimental results with other studies analyzing the role of (models of) different brain regions in processing texture information (e.g. Freeman&Simoncelli 11). Connections to models of pooling and computations in different visual areas should be matter of future research.

Reproducibility: No

Additional Feedback: # POST REBUTTAL After reading the reviews and author's reply, I keep my positive score: the authors have offered clarifications for the concerns and questions I had. I think is very interesting work and hope to see it in the conference. #

[Author Response · NeurIPS 2020]

We thank the reviewers for their detailed and useful reviews of our paper. We are glad the reviewers appreciated the interdisciplinary nature and the quality of our work. First, we recall that the main goal of the paper is to motivate and to provide a description of our interpolation method and to explain how it relates mathematically to other methods. Then, we illustrate how texture interpolation will serve further studies of visual perception. Importantly, our code is easy to use (from a command line) and will be available online so that the vision scientist community can start using it. Our future work will be dedicated to vision experiments *i.e.* directed toward a less theoretical audience. Future experiments (beyond this submission) should include a control with PS texture interpolation but not Gram-based texture interpolation (see §Patchy *vs* Non-patchy interpolation). If accepted, this paper will be the core technical reference.

**Balancing Methods and Results** As a compromise between R1 and R3, the method used to measure ellipticity will be moved to the supplementary material and expanded with more detailed explanation and an illustration. Specifically, prior to using our method we empirically validated it on artificial data with different dimensionality. R3 is correct that the distributions of natural images and textures are not elliptical. We only show that natural textures distributions are "more" elliptical than natural images distributions. Such a change and the extra-allowed page should leave some space to expand, and therefore clarify, the Results and Discussion sections and the description of our experiments as required by R1, R3 and R4.

**Figure importance and indexing** We will keep Figure 1/2 in the main paper because they provide intuition for why the proposed texture synthesis approach works. However, we will add a new Figure 1 illustrating the main idea of our paper which is to evaluate how moving along interpolation paths affects visual perception and neural activity.

**Experiments** As suggested by all reviewers, we will run our psychometric experiments with naive participants ($\sim 8$ more for each textures) for the camera-ready version if our submission is accepted. This will allow for a population analysis. However, collecting more neurophysiological data is uncertain because of the Covid situation which has delayed many ongoing experiments.

**Patchy *vs* Non-patchy interpolation** The PS algorithm and ours generate non-patchy interpolation contrary to the Gram-based interpolation. Patchy interpolation are less interesting for studying texture perception because neuronal receptive fields are localized and could therefore respond to the statistics of one of the two interpolated textures depending on patches location. Yet, the question of why the Gram-based interpolations are patchy is open. In particular, it is not due to an over-parametrization of the Gram method as both methods have $O(N^2)$ parameters (in fact, the Wasserstein method has only $N$ more parameters). We suggest that the mathematical foundation of our approach will enable further progress compared to the engineering nature of the PS algorithm.

**The perceptual scale** The inverse of the perceptual scale $f^{-1}$ linearizes the perception of a physical parameters (here the interpolation weight $t$). Following ideas that the visual cortex linearizes transformations [3], we believe that such a function is important to predict the path of the neural activity from the path of the stimuli.

**V1 *vs* V4** Previous work has shown that V4 is also sensitive to textures [5, 6] and therefore makes it interesting to compare to V1. We will keep in mind R1's remarks for future experiments.

**Biological relevance of our approach** We acknowledge that, in principle, there might exist a simpler approach that captures structure of textures in a way that is closer to biological perception, as mentioned by R2. Yet, CNNs-based approaches are meaningful for at least three reasons: (i) a body of literature shows that CNN activations are able to linearly predict neural activity along the hierarchy of the visual cortex [8]; (ii) mixture of elliptical distributions are a promising model of CNN activations [7] and (iii) mixture of elliptical distributions account for neuron responses to natural images in V1 [1] (but are still to be tested for mid/high-level vision). Even if the CNN architecture is largely inspired by the visual cortex, we agree with R3's comment that CNN weights are not grounded in human perception. However, the brain is hypothesized to be adapted to the statistics of its natural environment which are reflected in CNN activations.

**Miscellaneous** We will include all suggested references in the introduction or the discussion. We will illustrate the sample diversity of our approach by adding multiple synthesis results from different random seeds in the supplementary material. We did not account for the power spectrum in our loss [4], and we acknowledge that this would be more rigorous. We will add this feature to our code. The effect of the number of VGG layers used to constrain the synthesis is similar to what is already known [2]: deeper layers account for larger spatial structures. To our knowledge, there is no comparison of the Gram *vs* Wasserstein loss. The reason is that, differences are not visible at first sight on the synthesized textures without sampling interpolation of textures. Such differences may be crucial for visual perception studies but less for computer graphics.

**References** [1] R. Coen-Cagli, A. Kohn, and O. Schwartz. "Flexible gating of contextual influences in natural vision". In: *Nature Neuroscience* (2015). [2] L. Gatys, A. S. Ecker, and M. Bethge. "Texture synthesis using convolutional neural networks". In: *Advances in Neural Information Processing Systems*. 2015. [3] O. J. Hénaff, R. L. Goris, and E. P. Simoncelli. "Perceptual straightening of natural videos". In: *Nature neuroscience* (2019). [4] G. Liu, Y. Gousseau, and G.-S. Xia. "Texture synthesis through convolutional neural networks and spectrum constraints". In: *2016 23rd International Conference on Pattern Recognition (ICPR)*. IEEE. 2016. [5] G. Okazawa, S. Tajima, and H. Komatsu. "Image statistics underlying natural texture selectivity of neurons in macaque V4". In: *Proceedings of the National Academy of Sciences* (2015). [6] G. Okazawa, S. Tajima, and H. Komatsu. "Gradual development of visual texture-selective properties between macaque areas V2 and V4". In: *Cerebral Cortex* (2017). [7] L. G. Sanchez-Giraldo, M. N. U. Laskar, and O. Schwartz. "Normalization and pooling in hierarchical models of natural images". In: *Current opinion in neurobiology* (2019). [8] M. Schrimpf et al. "Brain-Score: Which Artificial Neural Network for Object Recognition is most Brain-Like?" In: *bioRxiv preprint* (2018).


[Meta-Review · NeurIPS 2020]

Four knowledgeable referees support acceptance for the contributions, notably the proposed geodesic distance between elliptically symmetric distributions over CNN filter outputs as a way to interpolate between textures and the evaluation of the method by psychophysical similarity and neural responses. I also recommend acceptance. However, please consider revising your manuscript– being more forthright in discussing the psychophysical results in comparison to other studies and also the non-ellipticalness of natural textures as pointed out by a reviewer.